# Deciphering the Role of the Anaphylatoxin C3a: A Key Function in Modulating the Tumor Microenvironment

**DOI:** 10.3390/cancers15112986

**Published:** 2023-05-30

**Authors:** Jolimar Hanna, Franck Ah-Pine, Chailas Boina, Yosra Bedoui, Philippe Gasque, Axelle Septembre-Malaterre

**Affiliations:** 1Unité de Recherche EPI (Études Pharmaco-Immunologiques), Université de La Réunion, Allée des Topazes, 97405 Saint-Denis, France; 2Laboratoire d’Immunologie Clinique et Expérimentale OI (LICE OI), CHU de La Réunion, Allée des Topazes, 97405 Saint-Denis, France; 3Service d’Anatomie et Cytologie Pathologiques, CHU de La Réunion, Avenue François Mitterrand BP450, 97448 Saint-Pierre, France

**Keywords:** tumor, anaphylatoxin C3a, complement, tumor microenvironment

## Abstract

**Simple Summary:**

The complement system is activated within the tumor microenvironment and its role in cancer development has raised much attention over the last decade. The aim of our study was to investigate the effects of the C3a anaphylatoxin on tumor cells (B16/F0 melanoma cell line) and two cell components of the tumor microenvironment: macrophages (Raw 264.7 Blue cell line) and mesenchymal stem cells (3T3-L1-like cell line). We showed that C3a plays a crucial role in the tumor microenvironment and may influence the tumor’s fate by modulating the expression of cytokines (including IL-10, TGFβ1), chemokines, Cox-2, and HO-1, and upregulating the oxidative stress response. We also demonstrated that C3a regulates VEGF expression, suggesting a role of the C3a/C3aR axis in angiogenesis. Our results provide novel insights into tumorigenesis and open new therapeutic avenues (C3aR antagonists) in cancer therapy.

**Abstract:**

The complement system plays a crucial role in cancer development. Our study investigated the role of C3a anaphylatoxin on the tumor microenvironment. Our models consisted of mesenchymal stem cells (MSC-like, 3T3-L1), macrophages (Raw 264.7 Blue, (RB)) and tumor cells (melanoma B16/F0). Recombinant mouse (Mo) C3a (rC3a) was produced in CHO cells transfected with a Mo-IL10-signal peptide-Mo C3a plasmid construct. The effects of rC3a, IFN-γ, TGF-β1, and LPS were tested on the expression of C3, C3aR, PI3K, cytokines, chemokines, transcription factors, antioxidant defense mechanisms, angiogenesis and macrophage polarization (M1/M2). 3T3-L1 expressed the highest levels of C3, while C3aR was expressed more by RB. Interestingly, expression of C3/3T3-L1 and C3aR/RB was markedly upregulated by IFN-γ. rC3a was found to upregulate the expression of anti-inflammatory cytokines (IL-10) on 3T3-L1 and TGF-β1 on RB. rC3a also upregulated the expression of pro-inflammatory cytokines in RB. The expression of CCL-5 increased in 3T3-L1 in response to rC3a. On RB, rC3a did not alter M1/M2 polarization but upregulated the expression of antioxidant defense genes, HO-1, and VEGF. C3/C3a produced mainly by MSC may play a critical role in TME remodeling by stimulating both anti-inflammatory and proangiogenic activities of tumor stromal cells.

## 1. Introduction

The complement system is an integral and highly regulated part of the host immune system [1,2]. It bridges innate and the adaptative immunity [3], leads to the removal of pathogens, complements the inflammatory processes (by triggering the release of inflammatory cytokines), and plays a critical role in the responses to different diseases and infections [4]. This system consists of a group of soluble and membrane-bound components (serum proteins, inhibitors, cell surface receptors, and regulators) that can be activated in a cascade like-fashion through three different pathways [5]: the classical pathway (CP) [6], the alternative pathway (AP), and the lectin pathway (LP). Each of these pathways can be triggered by a specific stimulus and converge on the cleavage of C3 by C3-convertase (either C2a4b (LP and CP) or C3bBb (AP)) that leads to the production of the anaphylatoxin C3a and the opsonin C3b. C3b continues the cascade until the formation of the anaphylatoxin C5a and the membrane attack complex (MAC; C5b-9) that helps eliminate pathogens through cytolytic activities [7].

The role of the complement in cancer is complex [8,9,10]. Many early studies described this system as protective from neoplastic diseases, mainly through complement-dependent cytotoxicity [11], antibody-dependent cell-mediated NK cytotoxicity [12], and/or membrane attack complex activity. Recently, a new paradigm has emerged: it was discovered that the complement had oncogenic capabilities. Indeed, many of its components were dysregulated in cancer cells and released in the tumor microenvironment (TME) [13], eventually leading to tumor progression, angiogenesis and favoring the tumor immune escape. Indeed, anti-tumor immunity may be regulated through the action of C5a and C3a and profit from the chronic inflammation sustained by these anaphylatoxins [14].

The anaphylatoxin C3a was always known for its pro-inflammatory function in stimulating mast cell degranulation and immune cell recruitment [15,16,17]. However, a new anti-inflammatory role has recently emerged [18,19]. Upregulated C3a levels have been found in many types of cancer, such as breast, colorectal esophageal, and brain cancers [20,21]. Additionally, in a model of breast cancer, it was shown that C3a promotes lung metastasis by activating cancer-associated fibroblasts (CAFs) [22].

Considering that the anaphylatoxin’s receptor C3aR (a G-protein coupled seven-transmembrane receptor) is present in a wide range of immune and non-immune cells (such as neurons, glial cells, [23,24] T cells, neutrophils, monocytes/macrophages, eosinophils, basophils, mesenchymal stem cells [25,26], and endothelial and epithelial cells) and that C3 can be generated locally [27], it was proposed that C3a plays a critical role in modulating TME. Thus, it has been demonstrated that the C3a-C3aR axis promotes breast cancer metastasis by reshaping the TME [22] and the invasion and migration of hepatocellular carcinoma cells [28]. Moreover, it has recently been shown that there is a strong link between high levels of VEGF^+^ and C3aR^+^ expressed by TAM in several models of cancer [21,29,30,31]. Indeed, using WT and C3aR^−/−^ animals, a unique subset of F4/80 ^high^ C3aR ^high^ TAM was identified as driving robust angiogenesis and metastasis [29]. Using the C3aR^−/−^ mice, it was already known that C3a controls the innate and adaptive immune responses, for instance, in the mouse B16 melanoma model [32]. The growth of B16 was retarded in the KO, together with an alteration in the levels of infiltrating immune cells. While the number of neutrophils and CD4 lymphocytes was increased, macrophages (TAM) were reduced in the TME [32].

We sought to better evaluate the role of C3a in the modulation of the TME and its pro-tumorigenic activities. Hence, we evaluated how C3a acts on three types of cells involved in the immune response and tumor development, namely macrophages (RAW 264.7 BLUE, RB), tumor cells (melanoma B16/F0) and mesenchymal stromal cells (MSC) (3T3-L1). In this study, we were interested in deciphering the regulated expression of C3aR, angiogenic factors, and molecules involved in inflammation, and the macrophage polarization, under the influence of C3a.

## 2. Materials and Methods

### 2.1. Cell Lines and Reagents

CHO, 3T3-L1, RAW 264.7 BLUE and B16/F0 cells were obtained from American Type Culture Collection (ATCC, Manassas, VA, USA). CHO cells were cultured in HAM/F12 medium, while 3T3-L1, Raw 264.7 Blue and B16/F0 cells were cultured in Dulbecco’s Modified Eagle’s Medium (DMEM) supplemented with 25 mM glucose, 10% heat-inactivated fetal bovine serum (FBS, PAN Biotech, Aidenbach, Germany, 3302 P290907), L-glutamine (2 mM Biochrom AG, Berlin, Germany, K0282), 0,1 mg/mL penicillin–streptomycin (PAN Biotech, P0607100), 1 mM of sodium pyruvate (PAN Biotech, Aidenbach, Germany, P0443100) and 0.5 µg/mL of amphotericin B (PAN Biotech, Aidenbach, Germany P0601001).

Recombinant mouse IFN-γ and Human TGF-β1 (cross-active for mouse cells) were purchased from Peprotech, while LPS was obtained from Sigma (Darmstadt, Germany).

### 2.2. Cell Transfection

Chinese hamster ovary (CHO) and mouse Raw 264.7 Blue cells were transfected with the pcDNA.3 plasmid containing the Mouse IL-10 signal peptide (sp)-Mo C3a sequence, synthesized and cloned by Genecust. We used the Lipofectamine kit (Ref L 3000008), as recommended by the supplier (Invitrogen, Waltham, MA, USA), and cells were selected with hygromycin (Ref 10687010, Invitrogen) at a concentration of 400 µg/mL for 7–10 days. CHO clones expressing C3a were obtained by limiting dilutions in 96-well plates.

### 2.3. Cell Culture and Stimulation

Cells were placed in 6-, 24-, or 96-well plates and maintained at 37 °C in a humid atmosphere with 5% CO_2_. The medium was replaced twice a week until 80–90% confluency. Cells were treated with either IFN-γ (20 ng/mL), TGF-β1 (20 ng/mL), LPS (1 µg/mL) CHO C3a F3 clone, or control (non-transfected) CHO supernatants (2% and 20%) for 24 h at 37 °C in a humid atmosphere with 5% CO_2_.

### 2.4. Cytotoxicity Assay

To investigate cytotoxicity, cells were treated for 24 h with either IFN-γ (20 ng/mL), TGF-β1 (20 ng/mL), LPS (1 µg/mL), CHO C3a F3 clone or control CHO supernatants (2% and 20%). Lactate dehydrogenase (LDH) levels released by lysed cells in the culture medium were measured using a colorimetric-based kit (ref. G1781, CytoTox 96^®^ Non-Radioactive Cytotoxicity Assay, Promega, Madison, WI, USA). We evaluated cytotoxicity as the ratio of released LDH levels after treatments compared to the maximum LDH release induced by Triton 1% exposure.

We also evaluated cell mitochondrial metabolic activity, performing an MTT (3-(4-5-dimethylthiazol-2-yl)-2,5-diphenyltetrazolium bromide) assay, according to the method of Mosmann [33]. Cells were cultured (4 × 10^3^ cells/well) in 96-well plates for 24 h before medium removal and treatment with either IFN-γ (20 ng/mL), TGF-β1 (20 ng/mL), LPS (1 µg/mL), CHO C3a F3 clone or control CHO supernatants (2% and 20%) for 19 h. The culture medium was then supplemented with 20 µL of sterile filtered MTT solution (5 mg/mL in phosphate-buffered saline (PBS)) (Sigma-Aldrich, Darmstadt, Germany) for 5 h. Then, the culture medium was removed and formazan crystals were solubilized in 200 µL of dimethyl sulfoxide. Finally, the absorbance was read at 560 nm (BIOTEK Cytation 5 imaging reader).

### 2.5. qRT-PCR (Sybergreen) Analyses

Total RNA extraction from harvested cell cultures (6-well plates) was performed using a Zymo kit (ZYMO, Catalog R1035). Extracted RNA was stabilized in 200 µL of RNA Shield and 800 μL of lysis buffer, collected and kept at −20 °C. qRT-PCR was performed using the One Step Bioline Sensifast Probe NO-ROX One step Kit (Meridian Bioscience, Cincinati, OH, USA, Bio-76005) and the Syber Green reagent (Lonza, Rockville, MD, USA, Cat.No.50513). The reaction mix contained 1 μL of extracted total RNA, 1.3 μL of primers mix (final primer concentration of 250 nM), and 2.7 μL of enzyme mix. qRT-PCR was performed using a Quantstudio 5 PCR thermocycler (Thermo Fisher Scientific, Waltham, MA, USA). The 2−ΔΔCT method has been used as a relative quantification strategy for quantitative real-time polymerase chain reaction data analysis. Moreover, GAPDH was selected as the reference gene to calculate relative gene expressions. Experiments were performed in triplicate. Primer and probe sequences are listed in Table 1.

### 2.6. Immunofluorescence Staining and Microscopy

Cells were cultured on glass coverslips in a 24-well plate and treated with IFN-γ (20 ng/mL), TGF-β1 (20 ng/mL), LPS (1 µg/mL) or CHO F3 clone supernatants (2% and 20%) for 24h. Cells were fixed and permeabilized in 99% ethanol at room temperature for 20 min, then incubated overnight with primary anti-mouse C3a antibodies (dilution 1/500) (BD Biosciences, San Jose, CA, USA, Clone: 187-1162). Cells were subsequently incubated with Alexa Fluor^®^488-conjugated Goat Anti-Rat secondary antibody (Invitrogen, Thermo Fischer Scientific, Waltham, MA, USA). The nuclear was counterstained with the nuclear fluorochrome 4’,6-diamidino-2-phenylindole (DAPI). Finally, the coverslips were mounted with Vectashield (Vector Labs, Newark, CA, USA). Images were acquired using a ×60 objective on a Nikon Eclipse E2000-U microscope and a Hamamatsu ORCA-ER camera (NIS-Element BR 5.20.00 imaging software, Nikon, Tokyo, Japan).

### 2.7. Immunohistochemical Staining

A melanoma case and a benign case of naevocellular nevus (benign melanocytic tumor) were retrieved from the tumor tissue bank of the CHU of La Réunion (French government CRB registry: DC-2016-145 2860). The formalin-fixed paraffin-embedded (FFPE) tissues were cut into 4 μm-thick sections. One section was stained with Hematoxylin–Phloxine–Safran by a trained pathologist to confirm the diagnosis of melanoma. Serial FFPE sections were processed using the Leica Biosystems Bond-III automated staining system for immunohistochemical staining with the mouse anti-human C3a (clone 184 K13/16, Biolegend, San Diego, CA, USA),or the mouseanti-human C3aR (clone D12, ref SC133172, Santa Cruz, CA, USA), the anti-human CD163 (clone 10D6, Leica), the anti-human CD68 (clone 514H12, ready-to-use, Leica), the anti-human CD248 (clone E9Z7O, Cell Signaling, Danvers, MA, USA), and the anti-human SOX10 (clone EP268, ready-to-use, BIO SB). Alkaline phosphatase-based detection (red staining) was carried out using the BOND polymer refine detection kits (Ref Ds9390) according to the manufacturer’s instructions (Leica, Paris, France).

### 2.8. Statistical Analysis

Results were expressed as mean ± standard error of the mean (SEM). All experiments consisted of three independent biological replicates. Statistical analysis was performed with GraphPad Prism 6 software. Two-way ANOVA tests followed by Bonferroni multiple comparison tests were performed (significance *p* < 0.05).

## 3. Results

### 3.1. Cell Viability

We assessed the viability of 3T3-L1, B16/F0 and Raw 264.7 Blue cells to different sets of stimulation, such as IFN-γ (20 ng/mL), TGF-β1 (20 ng/mL), LPS (1 µg/mL) or CHO F3 clone supernatants (2% and 20%) for 24 h. Control (CT) cells were treated with control CHO supernatants (20%). LDH release assay revealed that there was no effect on cell viability compared to the basal release in control cells (Figure 1). In addition, MTT assay revealed that all treatments used did not affect mitochondrial metabolic activity.

### 3.2. C3/C3aR Expression

Next, we aimed to dissect the contribution of inflammatory mediators [34] during tumor development in regulating the expression of the C3 (precursor of C3a) and its receptor C3aR coupled to PI3K signaling. In addition, several pro-inflammatory mediators were tested given that they have been identified as playing a key role in proliferation, angiogenesis, tissue invasion, and metastasis propensity. They include TNF-α, Il-6, IL-1β, but also chemokines (IL-8 (CXCL8), CCL-2, CCL5) [35]. We stimulated all three cell types to study C3aR/C3 regulated expression (Figure 2). Indeed, inflammation occupies an important position in the establishment of the TME that is decisive for the progression of oncogenesis.

C3aR mRNA expression was observed in all three cell models, while the highest expression (by a factor of 100-fold/B16 melanoma) was detected in Raw 264.7 Blue cells. After 24 h of stimulation by IFN-γ (20 ng/mL), TGF-β1 (20 ng/mL), or LPS (1 µg/mL), Raw 264.7 Blue showed significantly higher expression of C3aR mRNA (Figure 2A). All treatments also upregulated the expression of C3aR mRNA on 3T3-L1.

C3 mRNA expression was more prominent in 3T3-L1 cells (50-60-fold/B16 melanoma) and was upregulated essentially in response to IFN-γ, a cytokine known to be released by intra-tumoral T cells (Figure 2B).

PI3K signaling pathway is known to play an essential role in macrophage polarization [36]. It was also demonstrated that C3a alters the function of cancer-associated fibroblasts (CAFs known as MSC in cancer) by binding to its receptor and activating the PI3K/AKT pathway [22]. Both B16/F0 and 3T3-L1 expressed the highest levels of PI3K mRNA and particularly in response to TGF-β1 treatment for 3T3-L1 (Figure 2C).

In Figure 2D, we observed a strong increase in IL-6 gene expression in all three cell types in response to IFN-γ. LPS had a pro-inflammatory action in 3T3-L1 and Raw Blue. LPS had a pro-inflammatory action in 3T3-L1 and Raw Blue. TNF-α gene expression was increased in response to LPS and IFN-γ treatment in all three cell types (Figure 2E). TGF-β was only active in 3T3-L1 cells. Regarding chemokine expression (Figure 2F,G) IFN-γ showed the highest proinflammatory activity in all three cell types.

### 3.3. Recombinant C3a in CHO Expression

Recombinant C3a (rC3a) can be produced in bacteria but may potentially be contaminated by LPS. Hence, we generated different clones of transfected CHO cell lines that express mouse rC3a (Figure 3). The transfection was performed using a Mouse-IL10sp-Mo-C3a plasmid construct (Figure 3A). After validating the efficiency of CHO transfection by qRT-PCR and the expression of the Mo-IL10sp-Mo-C3a mRNA (Figure 3B), we validated the expression of the C3a protein via immunofluorescence using a specific rat anti-mouse C3a antibody (in green) (Figure 3C). The supernatants from one of the CHO transfected clones obtained by limiting cell dilution, F3, cultured in DMEM in the absence of hygromycin, were used as a source of rC3a and tested on all three cell types at a percentage of 2% and 20%. Conditioned medium was prepared using the control CHO cells and stored until use.

### 3.4. Modulation of C3aR and PI3K Expression by Recombinant C3a

rC3a significantly enhanced the expression of its own receptor in Raw 264.7 Blue cells, 3T3-L1 and, to a much lesser extent, in B16/F0 melanoma (Figure 4). The stimulation by rC3a significantly increased the expression of C3aR ~4-fold in 3T3-L1 and ~5-fold in Raw264.7 Blue cells (Figure 4A). This result implies that there is a positive autoregulation loop exerted by rC3a. PI3K mRNA was significantly increased by ~4-fold in 3T3-L1 cells but there was no significant difference in Raw 264.7 Blue cells (Figure 4B). It is important to note that PI3K mRNA level does not reflect the status of its activated phosphorylated form.

### 3.5. Modulation of Cytokine Expression by Recombinant C3a

To further understand the immunoregulatory role of rC3a, we assessed the expression of genes associated with the regulation of inflammatory responses including TNF-α, NFκB, IL-1β, IL-6, IL-10 and TGF-β1 by qRT-PCR in all three cell types (Figure 5). Focusing on anti-inflammatory cytokines (associated with pro-tumoral activities), we found that IL-10 expression was more prominent on 3T3 and upregulated by rC3a. With regard to TGF-β1, known for its immune-regulatory and pro-fibrotic activities, it was more expressed by Raw 264.7 Blue cells, particularly in response to rC3a. Pro-inflammatory cytokines (e.g., TNF-α, IL-1β) and the associated transcription factor NFκB were mainly expressed by RAW cells and rC3a upregulated the expression of all three. Interestingly, the pleiotropic cytokine IL-6, mainly expressed by 3T3-L1, was downregulated by rC3a.

### 3.6. Modulation of Chemokine Expression by Recombinant C3a

In this experiment, we studied the effect of rC3a on the major chemokines involved in inflammatory status, such as CCL-2, CCL-5 and MSC stem cell function CXCL-12 (CXCL-12, also known as the SDF-1 alpha) (Figure 6). CCL-5 is a major chemoattractant for lymphocytes (particularly T cells) and we found that rC3a significantly increased the expression of CCL-5 by all three cell types, although the highest level of expression was found for 3T3-L1 (Figure 6A). The constitutive expression of CXCL-12 (Figure 6C) was also more prominent for 3T3-L1 MSC-like cells but with a downregulating activity of rC3a. CCL-2 is very modestly expressed by B16/F0 and Raw 264.7 Blue and its level of expression was upregulated by rC3a. rC3a, in contrast, had no effect on CCL-2 expression by 3T3-L1, which expressed the highest level of CCL-2 mRNA compared to Raw 264.7 Blue and B16/F0 (Figure 6B).

### 3.7. Regulation of COX-2 Expression Involved in Prostaglandin Biosynthesis by Recombinant C3a

Upregulation of Cox-2, a rate-limiting key enzyme in the synthesis of prostaglandins, plays an important role in the control of inflammation associated with cancer. We found that rC3a significantly increased the expression of COX-2 in B16/F0 and 3T3-L1 cells but had no effect on Raw 264.7 Blue cells (Figure 7).

### 3.8. Modulation of Antioxidant Defense System by Recombinant C3a

We examined the effects of rC3a on the regulation of the oxidative stress response, and measured RNA expression of SOD-1-2, catalase, and Nrf2. The upregulated expression of all antioxidant genes was particularly prominent in Raw 264.7 Blue cells (Figure 8). SOD-2 and catalase expression was also upregulated in B16/F0, while no major regulation of these genes was noted when C3a stimulated 3T3-L1 cells. The results indicated that C3a may participate in the regulation of oxidative stress (which plays a crucial role in tumorigenesis and metastasis) by increasing the expression of antioxidant systems in macrophages and to a less extent in tumor cells.

### 3.9. Pro-Angiogenic Role of C3a

Angiogenesis has an undeniable role in the aggressiveness of tumor cells and in metastasis [37]. Moreover, it was reported that in a model of ovarian cancer, C3 silencing in cancer cells reduced the microvessel density in tumors [38]. Here, we wanted to see the effect of C3a on VEGF expression, which has a crucial role in promoting angiogenesis (Figure 9). Interestingly, we found that rC3a modulates the angiogenic properties of macrophages (Raw 264.7 Blue), mesenchymal (3T3-L1) and tumor cells (B16/F0) by significantly increasing the expression of VEGF mRNA by these cells (Figure 9A). In addition, rC3a significantly enhances these cells’ expression of HO-1 (Figure 9B). It is important to note that high levels of HO-1 have been particularly associated with tumor-associated macrophages (TAMs) with a polarization program similar to the M2 tumor promoting macrophage phenotype [39].

### 3.10. Expression of C3a and C3aR in Human Melanoma

Lastly, we used immunohistochemistry to investigate the expression of C3a and C3aR in a case of human melanoma (Figure 10A–F) and in a benign case of human naevocelllular naevus (Figure 10G,H). In melanoma, both C3a and C3aR expressions predominated in the perivascular areas. C3aR+ cells had a macrophage-like morphology with an abundant cytoplasm and a small, round nucleus. Interestingly, we observed that C3a localized closely to C3aR+ cells within the perivascular bed (Figure 10E,F, arrowhead). In addition, we detected a small subset of tumor cells that express C3a, but not C3aR. Within the benign naevus, C3a and C3aR were not expressed (Figure 10G,H).

## 4. Discussion

A new function of complement in cancer has emerged, mainly through the action of the anaphylatoxins C3a/C5a and their capacity to modulate immune cell and MSC functions.

Clinical observations and animal studies showed the critical role of C3a in supporting tumor growth: C3 (and C3a cleavage product) produced by CD8 + T cells may promote tumor progression by inhibiting these cells’ IL-10 production [27].

In contrast, other reports found that mice lacking C3 or C5aR showed decreased tumor growth [14]. Additionally, cancer treated with C3a, C3aR and/or C5aR inhibitors showed less resistance to therapy or had restricted metastasis [29,32,40], arguing that C3a may have pro-tumorigenic activities, yet poorly characterized mechanisms.

Here, we demonstrated that different stimulatory paradigms present in the stroma of tumors and mobilized either via innate immune PRRs (i.e., TLR4 agonist LPS), adaptive immunity (i.e., IFN-γ, produced by T cells) and via tumor growth factor (exemplified by profibrotic TGF-β1), regulated the expression of C3 as well as C3aR and downstream PI3K signaling molecule.

Most of all, we found that the C3a/C3aR axis involved principally and, respectively, 3T3-L1 on one hand and Raw 264.7 Blue cells on the other, arguing for the capacity of MSC-derived C3a to influence macrophage behavior via C3aR.

Unexpectedly, we found that IFN-γ, a cytokine expressed by infiltrating tumoral T cells and which is generally thought to protect against tumor expansion, markedly increased both C3 (precursor of C3a) and C3aR expression levels. It will be important to better address the phenotype of infiltrating T cells producing IFN-γ which may be differentiating into immunoregulatory Th2 or Treg phenotypes while infiltrating tumor beds. The capacity of C3a to upregulate the expression of TGF-β1 and IL-10, respectively, by Raw 264.7 Blue cells and 3T3-L1 is interesting and may represent a plausible link between C3a, polarized Treg differentiation, and immunoparalysis, given that both cytokines control lymphocyte adaptative immune functions. T cells infiltrate tumors in response to chemokines and, indeed, we found that C3a was able to upregulate CCL-5 but not CCL-2 (macrophage chemokine) via 3T3-L1 cells. CCL-5 is a chemokine involved in T cell chemotaxis. We also tested but failed to validate the capacity of C3a to regulate the expression of the chemokine CXCL-12, also known as a stromal growth factor (SDF-1).

Focusing on macrophages, it has been reported that C3a controls TAM’s polarization through C3a-C3aR-PI3Kγ signaling pathway. Mechanistically, it was shown that C3a could modulate TAMs into an M2 phenotype through the C3a-C3aR-PI3Kγ pathway [41]. Using our Raw cell model, we were not able to reach conclusive interpretations regarding the capacity of rC3a to modulate M2 versus M1 phenotypes. Our study was limited to a subset of cell markers, and a more comprehensive study needs to be performed. However, our data clearly indicated that C3a upregulated the expression of two major proinflammatory cytokines, namely TNF-α and IL-1β. This result has to be correlated with increased expression of their transcription factor, NFκB, due to rC3a exposure. High reactive oxygen species (ROS) levels play an important role as a cancer stimulator, inducing a wide range of mutations through DNA damage. Oxidative stress occurs in cells when the generation of ROS overwhelms the cell natural antioxidant defenses. We found that rC3a enhanced intracellular ROS generation in Raw 264.7 Blue cells but this was accompanied by elevated expression of anti-oxidant defense molecules such as SOD1/2, and catalase. We also found that rC3a enhanced the expression of heme-detoxification molecule HO-1 in Raw 264.7 Blue macrophages.

The C3a-C3aR-PI3K pathway was demonstrated to have an important role in activating CAFs (representing the population of intra-tumoral MSC) leading to lung metastasis in a breast cancer model [22]. Angiogenesis is a key event in cancer progression [42]. It is highly regulated by cancer and stromal cells.

The role of anaphylatoxins in angiogenesis is poorly studied. C3 has shown an anti-angiogenic role in a model of retinopathy of prematurity (ROP): it was found that mice deficient in C3 displayed increased neovascularization in this model and in the in vivo Matrigel plug assay [43]. In contrast, both C3a and C5a promote choroidal neovascularization in an age-related macular degeneration. In this study, we demonstrated that C3a affects angiogenesis by regulating the essential vascular endothelial growth factor (VEGF) expression by 3T3-L1 cells. It will be important to ascertain whether the C3a-C3aR axis may also regulate the production of other pro-angiogenic factors. We and others have recently found a strong link between high levels of C3aR+ TAM and VEGF expression in human and mouse models of cancer [21,29]. Indeed, Davidson and colleagues reported that the lack of a functional C3aR leads to a reduction in the B16 tumor mass and is associated with an increase in non-differentiated pro-inflammatory monocytes [29]. These findings support the critical role of C3a in monocyte recruitment and differentiation to anti-inflammatory macrophages. C3aR expression was also confirmed in the study that used the MN/MCA1 fibrosarcoma or 3-MCA-derived tumor models. The authors demonstrated that C3aR was mainly expressed by TAM [31]. These authors reported an upregulation of M1-like markers (CD11c, major histocompatibility complex class II (MHC-II), CD80, and CD86) and a reduction in the frequency of CD206+ (an M2-like marker) TAMs in C3aR-deficient mice. The frequency of infiltrating CD4+ T lymphocytes was significantly increased in C3aR-deficient mice to mediate a more robust adaptive immune response against cancer [31] and in light of the observation that the number of CD8+ infiltrating T cells was also higher in C3aR KO [29]. Thus, our results using model cell lines and the aforementioned mouse C3aR KO models argue for a major capacity of C3a/C3aR signaling to control immune cell infiltration and anti-tumor responses.

Lastly, we showed through immunohistochemistry that C3a and C3aR were mainly expressed and colocalized within the perivascular areas in a human melanoma tumor case. C3a and C3aR were totally absent from our benign case of human naevocelllular naevus. The perivascular preferential localization of both markers in melanoma supports our in vitro results arguing for a potential role of C3a/C3aR axis in angiogenesis and with the critical role of TAM in this process. Yet, the exact phenotype of those C3a+/C3aR+ perivascular cells is still to be demonstrated, in spite of their macrophage-like morphology. In addition, we observed that a subset of melanoma cells expresses C3a, suggesting a potential role of tumor cells in macrophage recruitment within the tumor microenvironment. Further morphological investigations are required to better characterize C3a and C3aR expressions in a much larger panel of human melanomas and to identify the differential phenotype of the C3a+ and C3aR+ cells.

## 5. Conclusions

In the present study, we show that the anaphylatoxin C3a plays a crucial role in the modulation of TME. It exerts a pro-(including the chemotaxis of T cells) and anti-inflammatory action depending on the cell types. These two effects may be interconnected in the case of melanoma (B16/F0) and transform TME into an immunosuppressive setting. A better understanding of the mechanism of action, the signaling pathways involved in modulating the TME with the identification of antagonists, will provide new breakthroughs in cancer therapy.

## Figures and Tables

**Figure 1 cancers-15-02986-f001:**
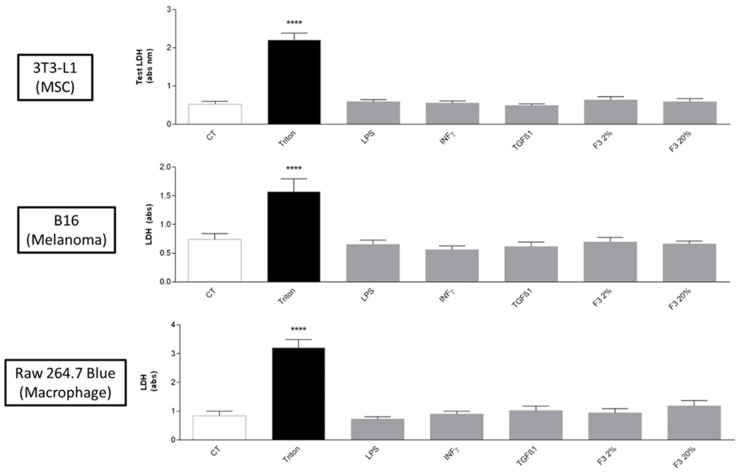
3T3-L1 (MSC), B16/F0 (Melanoma) and Raw 264.7 Blue (Macrophage) cells sensitivity and cytotoxicity assessment to different set of treatment. 3T3-L1, B16/F0/F0 and RAW 264.7 BLUE cells were stimulated with either IFN-γ (20 ng/mL), TGF-β1 (20 ng/mL), LPS (1 µg/mL), C3a F3 clone or control (CT) CHO supernatants (2% and 20%) for 24 h. Lactate dehydrogenase (LDH) levels released by lysed cells in the culture medium were measured using a colorimetric-based kit (ref. G1781, CytoTox 96^®^ Non-Radioactive Cytotoxicity Assay, Promega). We evaluated cytotoxicity as the ratio of released LDH levels after treatments compared to the maximum LDH release induced by Triton 1% exposure. Values correspond to means ± SEM of three independent replicates. Two-way ANOVA followed by a multiple comparison test (Bonferroni’s test) were performed. (**** *p* < 0.0001 as compared to non-stimulated cells (CT)).

**Figure 2 cancers-15-02986-f002:**
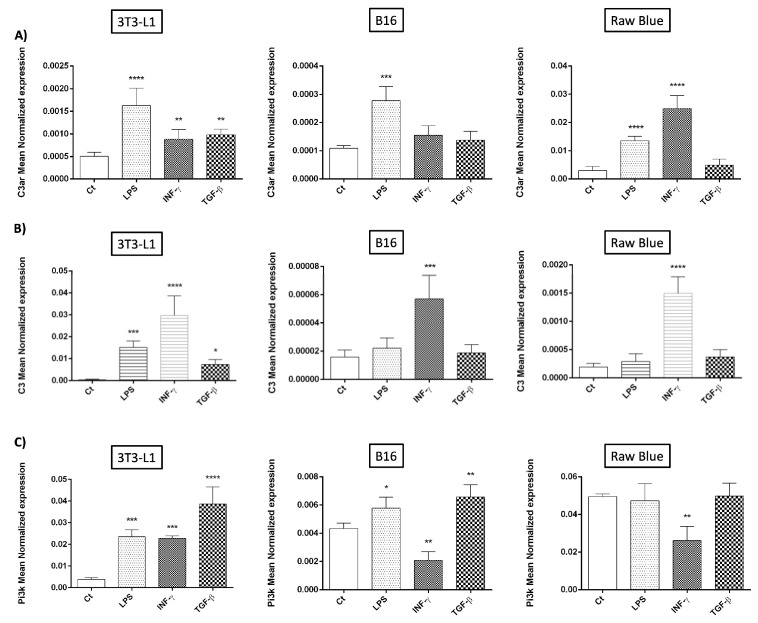
Different models of inflammation regulate C3aR, C3a and PI3K expression (at the RNA level). 3T3-L1, B16/F0 and Raw 264.7 Blue cells were stimulated with IFN-γ (20 ng/mL), TGF-β (20 ng/mL) and LPS (1 µg/mL) for 24 h. (**A**) C3aR, (**B**) C3a, (**C**) PI3K, (**D**) IL-6, (**E**) TNF-α, (**F**) CCL-2 and (**G**) CCL-5 RNA expression was determined by qRT-PCR (*n* = 3). Reported values are means ± SEM of three independent experiments. Two-way ANOVA followed by a multiple comparison test (Bonnferroni) was used to calculate the *p* value. (* *p* < 0.05, ** *p* < 0.01, *** *p* < 0.001 and **** *p* < 0.0001 compared to unstimulated cells (Ct)).

**Figure 3 cancers-15-02986-f003:**
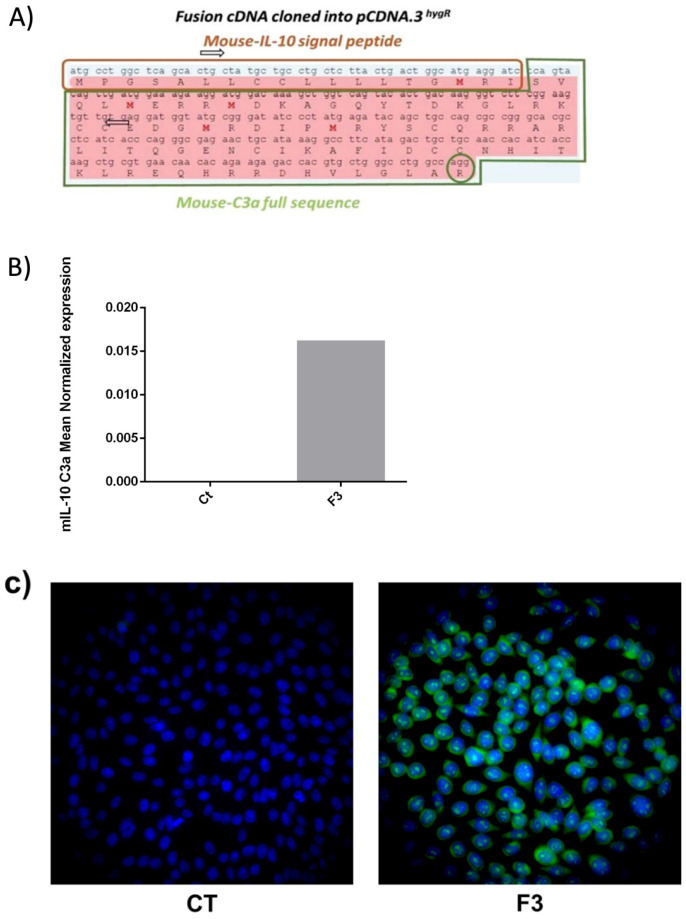
Recombinant mouse C3a expression from CHO transfected cells. (**A**) Schematic Representation of Mouse-IL10sp-Mo C3a plasmid construction including the position of the two primers used for RT-PCR. The terminal arginine (R) of C3a is indicated. (**B**) Mouse IL-10sp-Mo C3a mRNA expression was analyzed by qRT-PCR in one (F3) clone of transfected CHO cells versus non-transfected (control, CT). (**C**) F3 clone of CHO (F3) or non-C3a transfected CHO (CT) tested by immunofluorescence using the rat monoclonal anti-C3a followed by Goat anti-Rat Alexa 488.

**Figure 4 cancers-15-02986-f004:**
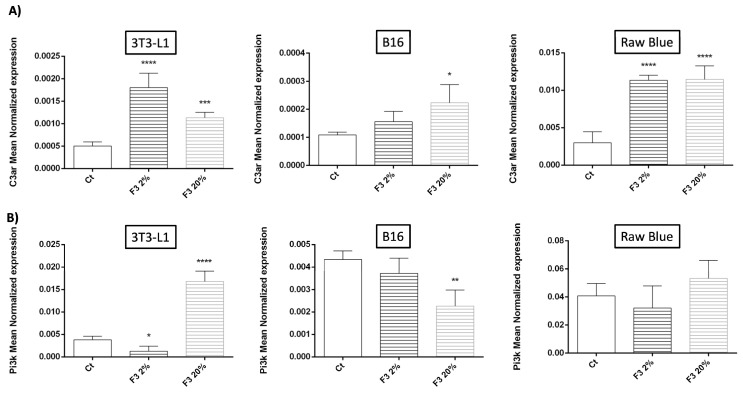
C3a modulates the expression of its own receptor and Pi3K. 3T3-L1, B16/F0 and Raw 264.7 Blue cells were stimulated with CHO F3 clone supernatants (2% and 20%) during 24 h. Ct cells are cells treated with conditioned medium from non-transfected CHO. (**A**) C3aR and (**B**) PI3K RNA expression were determined by qRT-PCR. Reported values are means ± SEM of three independent experiments. (* *p* < 0.05, ** *p* < 0.01, *** *p* < 0.001 and **** *p* < 0.0001 compared to non-stimulated cells (Ct)).

**Figure 5 cancers-15-02986-f005:**
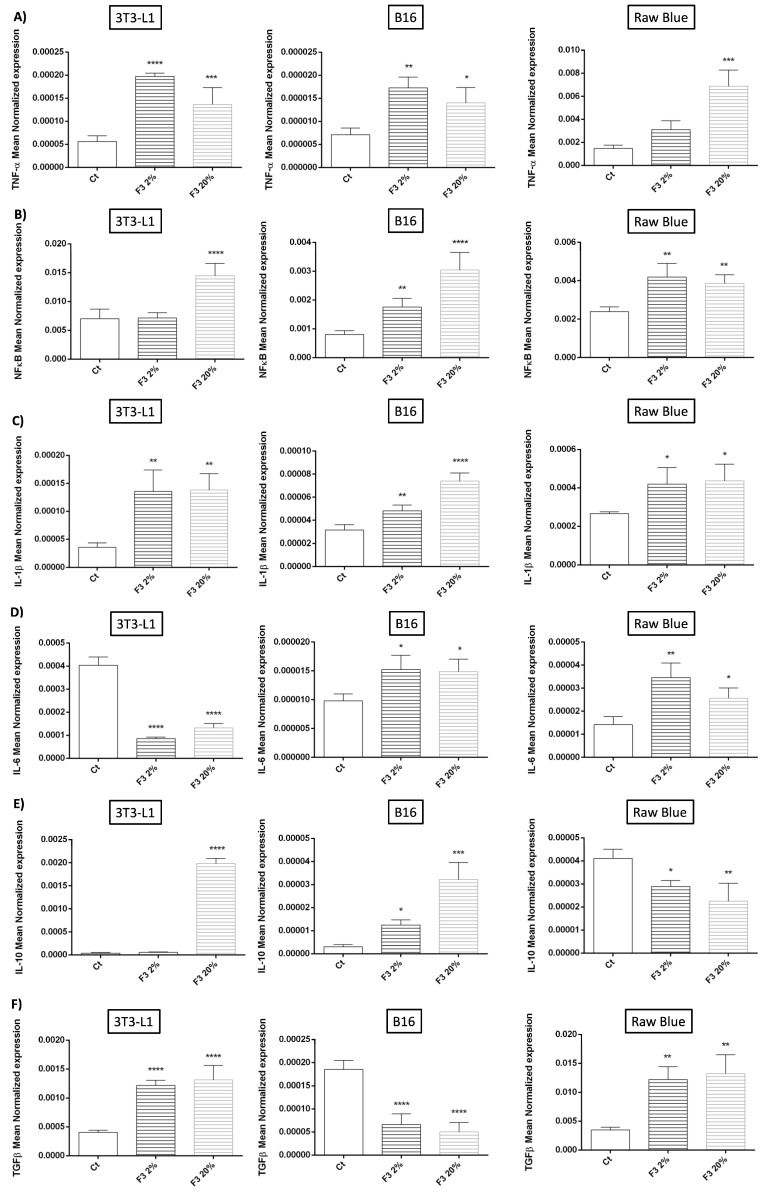
C3a immunoregulatory activities. 3T3-L1, B16/F0 and Raw 264.7 Blue cells were stimulated with CHO F3 clone supernatants (2% and 20%) during 24 h. Ct cells are cells treated with conditioned medium from non-transfected CHO. (**A**) TNF-α, (**B**) NF-κB, (**C**) IL-1β, (**D**) IL-6, (**E**) IL-10 and (**F**) TGF-β1, expression were determined by qRT-PCR. Reported values are means ± SEM of three independent experiments. (* *p* < 0.05, ** *p* < 0.01, *** *p* < 0.001 and **** *p* < 0.0001 as compared to untreated cells (Ct)).

**Figure 6 cancers-15-02986-f006:**
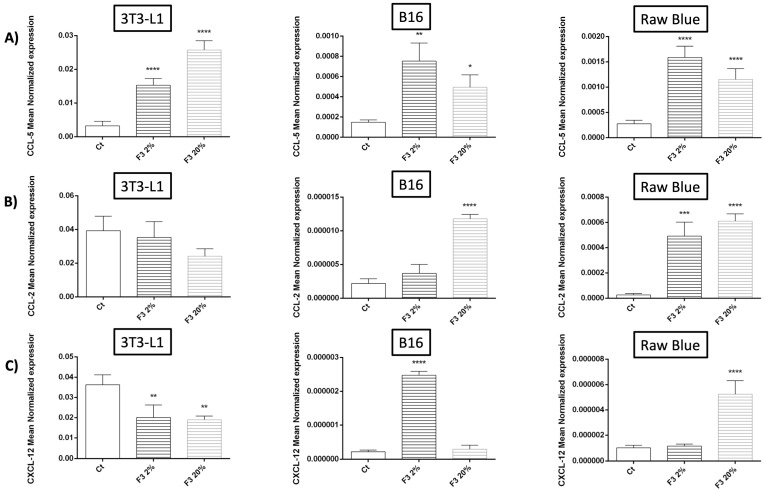
C3a differentially regulated the expression of the chemokines. 3T3-L1, B16/F0, and Raw 264.7 Blue cells were stimulated with CHO F3 clone supernatants (2% and 20%) for 24 h. Ct cells are cells treated with conditioned medium from non-transfected CHO. (**A**) CCL-5, (**B**) CCL-2, and (**C**) CXCL-12 expression were determined by qRT-PCR (*n* = 3). Reported values are means ± SEM of three independent experiments. (* *p* < 0.05, ** *p* < 0.01, *** *p* < 0.001 and **** *p* < 0.0001 as compared to untreated cells (Ct)).

**Figure 7 cancers-15-02986-f007:**
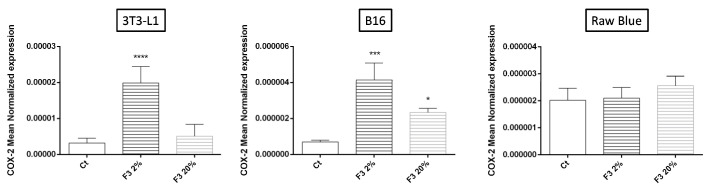
C3a effects on COX-2 in 3T3-L1, B16/F0 and Raw 264.7 Blue cells were stimulated with CHO F3 clone supernatants (2% and 20%) during a 24 h period. Ct cells are cells treated with conditioned medium from non-transfected CHO. COX-2 expression was determined by qRT-PCR. Reported values are means ± SEM of three independent experiments. (* *p* < 0.05, *** *p* < 0.001, **** *p* < 0.0001 as compared to untreated cells (Ct)).

**Figure 8 cancers-15-02986-f008:**
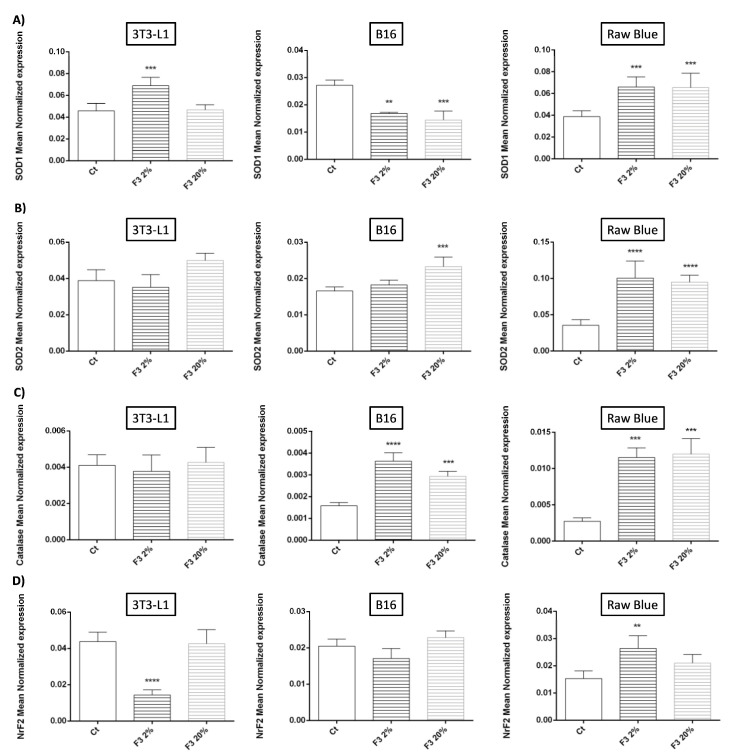
C3a enhances the antioxidants defense. 3T3-L1, B16/F0 and Raw 264.7 Blue cells were stimulated with CHO F3 clone supernatants (2% and 20%) during 24 h. Ct cells are cells treated with conditioned medium from non-transfected CHO. (**A**) SOD-1, (**B**) SOD-2, (**C**) Catalase and (**D**) Nrf2 expression were determined by qRT-PCR. Reported values are means ± SEM of three independent experiments. (** *p* < 0.01, *** *p* < 0.001 and **** *p* < 0.0001 compared to untreated cells (Ct)).

**Figure 9 cancers-15-02986-f009:**
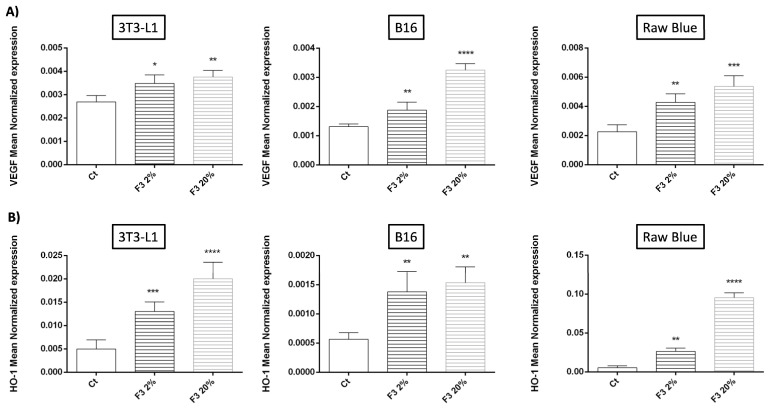
Pro-angiogenic effects of C3a. 3T3-L1, B16/F0 and Raw 264.7 Blue cells were stimulated with CHO F3 clone supernatants (2% and 20%) during 24 h. Ct cells are cells treated with conditioned medium from non-transfected CHO. (**A**) VEGF and (**B**) HO-1 expression were determined by qRT-PCR. Reported values are means ± SEM of three independent experiments. (* *p* < 0.05, ** *p* < 0.01, *** *p* < 0.001 and **** *p* < 0.0001 as compared to untreated cells (Ct)).

**Figure 10 cancers-15-02986-f010:**
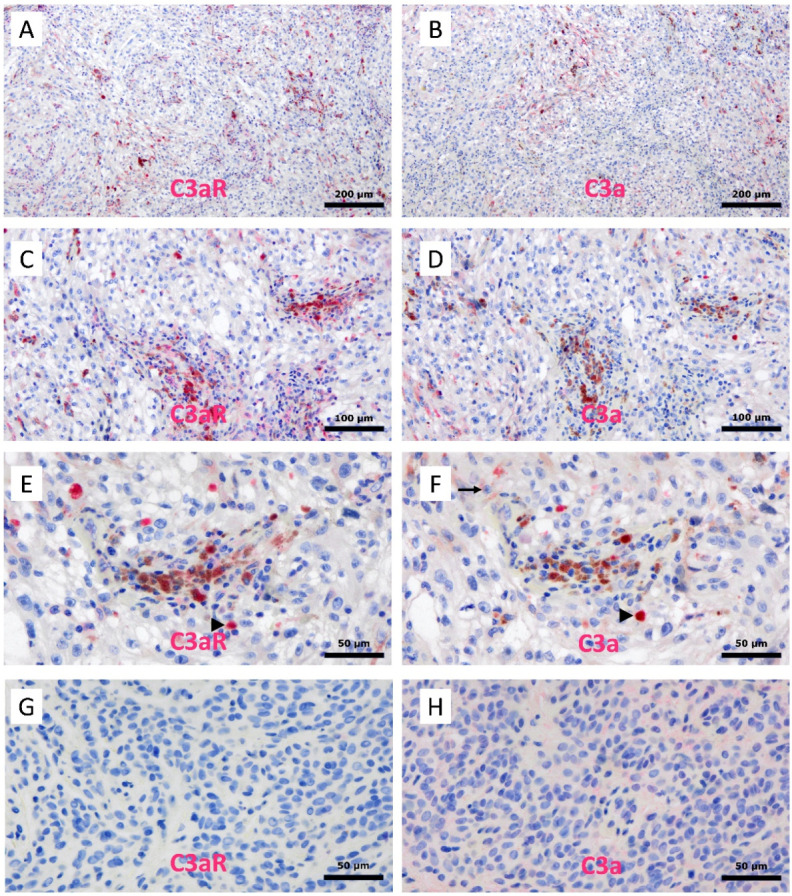
Immunohistochemical assessment of C3a and C3aR expression in human melanoma and benign naevocelllular naevus. Serial formalin-fixed paraffin embedded (FFPE) sections of a human melanoma (**A**–**F**) and a benign case of human naevocelllular naevus (**G**,**H**) were processed via alkaline phosphatase-based immunostaining with either C3aR (clone D12) (**A**,**C**,**E**,**G**) and C3a (clone K13/16) (**B**,**D**,**F**,**H**). Using serial tissue sections, we were able to identify cells single or double stained for both markers. For example, the arrow points to a C3aR+/C3a− cell and the arrowheads to C3aR+/C3a+ perivascular cells (macrophage-like) in melanoma. Brown background staining was detected due to the presence of the melanin pigment in melanoma. The benign naevus revealed no C3a/C3aR expression. Images are shown at ×100 (**A**,**B**), ×200 (**C**,**D**) and ×400 (**E**–**H**) magnification. In addition, we confirmed that, in malignant melanoma, the majority of C3a and C3aR cells co-localized with CD68 and CD163 macrophages within the perivascular areas (Figure 11A–D). In addition, few CD248/endosialin + MSC were observed within the vascular walls (Figure 11E). To a lesser extent, C3a cells colocalized with SOX10 (nuclear staining) melanoma tumor cells (Figure 11F). Notably, double-staining experiments—for example, those using peroxidase-conjugated antibody and DAB brown color development—were not conclusive due to the intrinsic presence of brown melanin pigment within the melanoma cells and melanophages.

**Figure 11 cancers-15-02986-f011:**
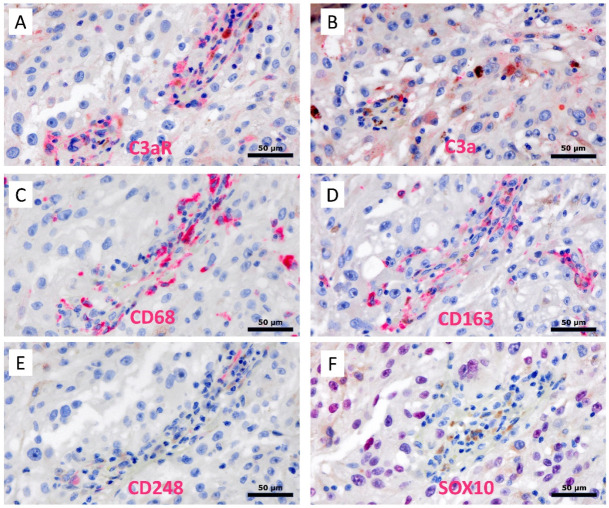
Abundant C3aR expression in perivascular regions of human melanoma. Serial FFPE sections of a human melanoma were stained using monoclonal antibodies for (**A**) C3aR, (**B**) C3a, (**C**) CD68, (**D**) CD163, (**E**) CD248, and (**F**) SOX10. Images are shown at ×400 magnification. The 50 μm size is indicated by the bar.

**Table 1 cancers-15-02986-t001:** Mouse Primers used for qRT-QPCR analyses.

Target Gene	Forward Sequence	Reverse Sequence	Detection
GAPDH	CGACTTCAACAGCAACTCCCACTCTTCC	TGGGTGGTCCAGGGTTTCTTACTCCTT	Sybergreen
C3aR	GATTTGTTGGTGGCTCGCAG	GAAACAGAGGCCGTGAGTGT	Sybergreen
VEGF	CTCCACCATGCCAAGTGGTC	GTCCACCAGGGTCTCAATCG	Sybergreen
NFκB-p50	CTCTGCATCAGTGACGGTAAAC	TTGTTGTTCTTCAGCCGTGC	Sybergreen
TNF-α	GATCGGTCCCCAAAGGGATG	CCACTTGGTGGTTTGTGAGTG	Sybergreen
CCL-2	CTTCTGGGCCTGCTGTTCAC	CTTGAGCTTGGTGACAAAAACTAC	Sybergreen
CCL-5	TGCCCTCACCATCATCCTCA	TTCCTTCGAGTGACAAACACGA	Sybergreen
IL-6	TGATGGATGCTACCAAACTGGA	TGTGACTCCAGCTTATCTCTTGG	Sybergreen
IL-10	GTAGAAGTGATGCCCCAGGC	CACCTTGGTCTTGGAGCTTATT	Sybergreen
HO-1	CACAGGGTGACAGAAGAGGC	CTGCAGGGGCAGTATCTTG	Sybergreen
TGFβ-1	CTGCTGACCCCCACTGATAC	GGGGCTGATCCCGTTGATT	Sybergreen
COX-2	GACACGACTTCGGAGGAGAG	AGACTTTGTCAGAAGTTCTTTTTGT	Sybergreen
SOD-1	CATGGCGATGAAAGCGGTG	GCACTGGTACAGCCTTGTGTA	Sybergreen
SOD-2	CACCGAGGAGAAGTACCACG	CTCCAGCAACTCTCCTTTGGG	Sybergreen
Nrf2 (Nfe2l2)	CCAGACAGACACCAGTGGAT	ATATCCAGGGCAAGCGACTCA	Sybergreen
Catalase	GTGCATGCATGACAACCAGG	GTGCATGCATGACAACCAGG	Sybergreen
PI3K (PRKAA1)	ACCATGGAGGAGAACCCTTATG	ACGGACAGTGCTCCTCCTTA	Sybergreen
CXCL-12	TGCTATGCTGCCTGCTCTTAC	CACAACACTTCCGAAGACCCT	Sybergreen
m-IL-10sp m C3a	GTCAAAATTTGCAACTATGTGGGG	GTTTGCAAAGCCAACCACCA	Sybergreen
IL1-β	TGCCACCTTTTGACAGTGATG	TGATGTGCTGCTGCGAGATT	Sybergreen
C3	GACAACAACCTACTGCCCGT	TTCCACTGCCCGATGTTGAC	Sybergreen

## Data Availability

Not applicable.

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
