# Peer review of "Deciphering the Role of the Anaphylatoxin C3a: A Key Function in Modulating the Tumor Microenvironment"

_cancers, 2023, doi:10.3390/cancers15112986_

Round 1

Reviewer 1 Report

The manuscript by Hanna et al. predominantly examines gene expression in cell lines that represent cell type found in the melanoma: B16F0 (tumor cells), 3T3-L1 (MSCs), and RAW264.7 Blue (macrophages).  They examined expression of complement protein C3a and its receptor C3aR in unstimulated and stimulated cells.  They then generated recombinant C3a in CHOs and examined the effect on gene expression of treating the 3 cell lines with conditioned media. Lastly the authors perform immunohistochemistry to localize C3a and C3aR in a primary melanoma tissue section.

Specific comments

1)     Should use gene names for the targets that are examined in RT-PCR assays.  This is particularly need for Nfkb and PI3k where there are several genes that code for subunits of NFkB and PI3K complexes but to be more precise should be used for Nrf2 where the gen name is Nfe2l2.

2)     A major problem is the wrong control was used for the RT-PCRs in figures 4-10 using 2% and 20% conditioned media from CHO-C3a cells.  If I interpreted the figure legend correctly it appears they use untreated cells as a control (ct).  It is more appropriate to use 2% and 20% conditioned media from unmodified CHOs.  The effect of other factors in the conditioned media other than the C3a may explain the frequent times when there was an effect observed with 2% media but not 20%. At a minimum the RT-PCRs which they highlight in the discussion IFN-γ markedly increasing C3 (precursor of C3a) and C3aR expression levels, upregulation of TGF-β1 and IL-10 respectively in Raw 264.7 Blue cells and 3T3-L1, and CCL-5 but not CCL-2 (macrophage chemokine) in 3T3-L1 cells.

3)     In figure 7 it appears the authors examined gene expression associated with M1 and M2 polarization after treating naïve macrophages with C3a conditioned media. It would be more informative if naïve macrophages were treated with C3a and then polarized with IFNg/ LPS (M1) and IL4/IL13 (M2) for 6-24h and then gene expression was examined.

4)     More information on how they report their RT-PCR data needs to be given. I assume that it is some version of ddCT comparing gene expression to Gapdh but the authors need to provide the formula they used in the material and methods section.

5)     Since the paper is examining C3a action on different cell types in the tumor microenvironment, the authors in figure 11 should perform immunohistochemistry to show localization of C3a and its receptor with macrophages, MSCs and tumor cell in the melanoma tissue sections.  Would also be interesting to compare to normal tissue if available (foreskin?).

Reviewer 2 Report

The authors present an interesting and well written report addressing the somewhat confusing and unresolved roles of complement and specifically Anaphylatoxin C3a.The research utilizes distinct mouse melanoma cells to understand how interactions with stromal cells such as bone marrow derived macrophages and mesenchymal stem cells can specifically promote tumor growth via Anaphylatoxin through the paracrine modification of cytokines and angiogenic growth factor VEGF. The authors go on to show how the results in a unique VEGF dependent phenotype with C3a signaling. Prior to publication I would suggest some significant considerations for the authors to enhance the information conveyed to their targeted audience. The greatest concern I have is the lack of other related works in the complement studies that led to this work, which needs to be amended to be more useful to readers.

Specific concerns:

1)    In figure 1  it might be helpful if the data is shown not normalized but the actualy observed measurements in absorbance I believe for this assay and then statisitical ANOVA to compare the two F3 treatments. It might also be helpful to label the three cell lines underneath with “msc, “melanoma” and “macrophage”.

2)    Table 1 lists the primers and has PI3K which I believe is the CA? It would be good if the authors confirm the species, mouse, human or Chinese hamster and that all sequences are correct.

3)    For measuring specific isoforms of PI3K I would expect that macrophages to have gamma or delta predominantly and that tumor cells have alpha or beta. Did the authors nuance these form of tumor and stroma PI3K signaling?

4)    In Figure 2 is there a control gene for inflammation such as IL-6 or TNFa that can be used to measure the general overall inflammatory response?

5)    In figure 3 did the authors check to see if the CHO cells changed their growth, morphology or general cellular behavior?

6)    The title and conclusions of figure 4 need to be carefully thought about what the experimental data shows and what the author claims this is showing.

7)    Did the authors also measure by ELISA the amount of cytokines produced in figures 5-7?

8)    In figure 7 a larger panel of M1 and M2 markers that have at least 4-5 for both would greatly enhance the findings.

9)    For figure 8-10 did the authors consider a positive or negative stimulation or inhibit of these respective pathways to see to what scale the relative gene expression changes from the conditioned medium was inducing or suppressing?

10) For figure 11 can the authors add something from larger datasets or more than one high powered representative images to appreciate the C3a/C3aR expression in melanoma?

11) I do not see a summary figure that was created with biorender as credited.

Fine, lacking more explanatory details in some instance.

Round 2

Reviewer 1 Report

The authors clarified questions regarding methodology from previous review satisfactorily. New data in figures 11 and 12 with primary tissue sections increase the impact of the paper.  Still have concerns about figure 7 and perhaps this is due to my misreading of the legend and Material and methods but it appears the authors are making conclusions about C3a’s effect on macrophage polarization without polarizing the cells.  If the authors don’t want to repeat the experiments by polarizing the cells to M1 (LPS and/or IFNg) and M2 (IL-4) then the data should just be removed from the paper because in its present form the data are not very informative.

Author Response

We would like to thank reviewer 1 for his comments and corrections on our manuscript.

We have removed figure 7.

Reviewer 2 Report

The authors have made significant improvements to their study, no further changes requested.

Author Response

Many thanks to reviewer 2.